# From *p*-Values to Posterior Probabilities of Null Hypotheses

**DOI:** 10.3390/e25040618

**Published:** 2023-04-06

**Authors:** Daiver Vélez Ramos, Luis R. Pericchi Guerra, María Eglée Pérez Hernández

**Affiliations:** 1Faculty of Business Administration, Statistical Institute and Computerized Information Systems, Río Piedras Campus, University of Puerto Rico, 15 AVE Universidad STE 1501, San Juan, PR 00925-2535, USA; 2Faculty of Natural Sciences, Department of Mathematics, Río Piedras Campus, University of Puerto Rico, 17 AVE Universidad STE 1701, San Juan, PR 00925-2537, USA; luis.pericchi@upr.edu (L.R.P.G.); maria.perez34@upr.edu (M.E.P.H.)

**Keywords:** *p*-value calibration, Bayes factor, linear model, pseudo-p-value, adaptive levels

## Abstract

Minimum Bayes factors are commonly used to transform two-sided *p*-values to lower bounds on the posterior probability of the null hypothesis, in particular the bound −e·p·log(p). This bound is easy to compute and explain; however, it does not behave as a Bayes factor. For example, it does not change with the sample size. This is a very serious defect, particularly for moderate to large sample sizes, which is precisely the situation in which *p*-values are the most problematic. In this article, we propose adjusting this minimum Bayes factor with the information to approximate an exact Bayes factor, not only when *p* is a *p*-value but also when *p* is a pseudo-*p*-value. Additionally, we develop a version of the adjustment for linear models using the recent refinement of the Prior-Based BIC.

## 1. Introduction

By now, it is well known by practitioners that *p*-values are not posterior probabilities of a null hypothesis, which is what science would need to declare a scientific finding. So *p*-values, and particularly the threshold of 0.05, need to be recalibrated. Two widespread practical attempts are (i) the so-called Robust Lower Bound on Bayes factors BF≥−e·p·log(p) [1] and (ii) the replacement of the ubiquitous α=0.05 by $\alpha^{*}=0.005$ [2]. These suggestions, which are an improvement of usual practice, fall short of being a real solution, mainly because the dependence of the evidence on the sample size is not considered. Still, the Robust Lower Bound is useful since it is valid from small sample sizes and onward and only depends on the *p*-value. It is known that the evidence of a *p*-value against a point null hypothesis depends on the sample size. In [3], they consider *p*-values in linear models and propose new monotonic minimum Bayes factors that depend on the sample size and converge to −e·p·log(p) as the sample size approaches infinity, which implies it is not consistent, as Bayes factors are. It turns out that the maximum evidence for an exact two-tailed *p*-value increases with decreasing sample size. There are several proposals in the literature, and most do not depend on the sample size, while those that do continue to be Robust Lower Bounds; however, neither behaves like a real Bayes factor. In this article, we propose to adjust the Robust Lower Bound −e·p·log(p) so that it behaves in a similar or approximate way to actual Bayes factors for any sample size. A further complication arises, however, when the null hypotheses are not simple, that is, when they depend on unknown nuisance parameters. In this situation, what is usually called *p*-values are only pseudo-*p*-values [4] (p. 397). So, we first need to extend the validity of the Robust Lower Bound to pseudo-*p*-values. The effect of adjusting this minimum Bayes factor with the sample size is shown in a simulation in Section 5.1.

The outline of the article is as follows: In Section 2 we define pseudo-*p*-values using the *p*-value definition of [4] (p. 397) and extend for them the validity of the Robust Lower Bound. In Section 3, we present the adaptive significance levels that will be used for incorporating the sample size in the lower bound: the general adaptive significance level presented in [5] and the refined version for linear models developed in [6]; in both cases, we use versions calibrated using the Prior-Based BIC (PBIC) [7]. In Section 4, we derive adaptive approximate Bayes factors and apply them to pseudo-*p*-values in Section 5. We close in Section 6 with some final comments.

## 2. Valid *p*-Values and Robust Lower Bound

Under the null hypotheses, *p*-values are well known to have Uniform(0, 1); in [4] (p. 397), a more general definition is given.

**Definition 1.** *A p-value* p(X) *is a statistic satisfying* 0≤p(x)≤1 *for every sample point **x***. Small values of p(X) *give evidence that* $H_{1}:\theta \in \Theta_{0}^{c}$ *is true, where* $\Theta_{0}$ *is some subset of the parameter space and* $\Theta_{0}^{c}$ *is its complement. A p-value is valid if, for every* $\theta \in \Theta_{0}$ *and every* 0≤α≤1,
$$P_{\theta }\left(p\left(X\right)\leq \alpha \right)\leq \alpha .$$

Based on this definition, we can say that there are valid *p*-values that are Uniformly Distributed in (0, 1), that is,
(1)$$P_{\theta }\left(p\left(X\right)\leq \alpha \right)=\alpha \;\;\mathrm{for}\;\mathrm{every}\;\;\theta \in \Theta_{0}\;\;\mathrm{and}\;\mathrm{every}\;0\leq \alpha \leq 1,$$
and others that are not, that is, when there is at least one α, such that
(2)$$P_{\theta }\left(p\left(X\right)\leq \alpha \right)<\alpha \;\;\mathrm{for}\;\mathrm{every}\;\;\theta \in \Theta_{0}.$$

**Remark 1.** 
*We consider any valid p-value complying with (Equation 2) a pseudo-p-value.*


The “Robust Lower Bound” (RLB), as we call it here and proposed by [1], is
$$B_{L}\left(p\right)=\left\{\begin{array}{cc} -e\cdot p\cdot \log(p) & p<e^{-1}\\ \;\;\;\;\;\;\;\;\;1 & \mathrm{otherwise} \end{array}\right.$$

The authors consider that under the null hypothesis, the distribution of the *p*-value, p(X), is Uniform(0, 1). Alternatives are typically developed by considering alternative models for X, but the results then end up being quite problem-specific. An attractive approach is instead to directly consider alternative distributions for *p* itself. In effect, they consider that, under $H_{1}$, the density of *p* is f(p|ξ), where ξ is an unknown parameter. So, consider testing
$$H_{0}:p∼\mathrm{Uniform}\left(0,1\right)\;\mathrm{versus}\;H_{1}:p∼f\left(p|\xi \right)$$

If the test statistic (T) has been appropriately chosen so that large values of T(X) would be evidence in favor of $H_{1}$, then the density of *p* under $H_{1}$ should be decreasing in *p*. A class of decreasing densities for *p* that is very easy to work with is the class of Beta(ξ,1) densities, for 0<ξ≤1, given by $f\left(p|\xi \right)=\xi p^{\xi -1}$. The uniform distribution (i.e., $H_{0}$) arises from the choice ξ=1 [1]. The expression $B_{L}\left(p\right)=\inf_{all\;\pi }B_{\pi }\left(p\right)$, where $B_{\pi }\left(p\right)$ is the Bayes factor of $H_{0}$ to $H_{1}$ for a given prior density π(ξ) on this alternative.

Note that this calibration has already been proposed in [8]. Another class of decreasing densities is Beta(1,ξ) with ξ>1. This leads to the “−e·q·log(q)” calibration, where q=1−p see [9].

In contrast with Remark 1, if we consider p(X) a pseudo-*p*-value under $H_{0}$, that is,
$$p∼\mathrm{Beta}\left(\xi_{0},1\right)\;\;\;\mathrm{with}\;\;\;\xi_{0}>1,\;\;\mathrm{fixed}\;\mathrm{but}\;\mathrm{arbitrary},$$
under the test
$$H_{0}:p∼\mathrm{Beta}\left(\xi_{0},1\right)\;\;\;\mathrm{vs}.\;\;\;H_{1}:p∼f\left(p|\xi \right)$$
with f(p|ξ)∼Beta(ξ,1) for $0<\xi \leq \xi_{0}$, then a generalized Robust Lower Bound $RLB_{\xi_{0}}$ can be defined as
(3)$$B_{L}\left(p,\xi_{0}\right)=\left\{\begin{array}{cc} -e\cdot \xi_{0}\cdot p^{\xi_{0}}\log\left(p\right) & p<e^{-\frac{1}{\xi_{0}}}\\ \;\;\;\;\;\;\;\;\;1 & \mathrm{otherwise} \end{array}\right.$$
where $\xi_{0}$ has to be estimated or calculated theoretically (see [10] for a proposal when extending for multiple testing). Any value $\xi_{0}\neq 1$ corresponds to a pseudo-*p*-value.

On the other hand, since $f\left(p|\xi \right)=\xi p^{\xi -1}$ has its maximum in $\xi =-\frac{1}{\log(p)}<1$ with $p<e^{-1}$, then f(p|ξ) is decreasing for $\xi >-\frac{1}{\log(p)}$, thus for any Bayes factor $B_{01}$
(4)$$B_{01}\geq B_{L}\left(p\right)>B_{L}\left(p,\xi_{0}\right)\;\;\;\mathrm{with}\;\;\;\xi_{0}>1$$
See Figure 1.

In the following, we calibrate $RLB_{\xi_{0}}$ such that $RLB_{\xi_{0}}\approx B_{01}$.

**Lemma** **1.** 
*$B_{L}\left(p_{val},\xi \right)=-e\cdot \xi \cdot p_{val}^{\xi }\cdot \log\left(p_{val}\right)\geq e\cdot \xi \cdot p_{val}^{\xi }>p_{val}^{\xi },$   for, $0<p_{val}<e^{-1}$ and ξ≥1. Note that $B_{L}\left(p_{val},1\right)=B_{L}\left(p_{val}\right)$*


**Proof.** Appendix A.    □

**Theorem 1.** *The* ${\mathrm{RLB}}_{\xi }$ *is a valid p-value for* ξ≥1, *that is*,
$$P(B_{L}\left(p,\xi \right)\leq \alpha |p∼f\left(p|\xi \right))\leq \alpha ,\;\mathrm{for}\;\mathrm{each}\;0\leq \alpha \leq 1.$$

**Proof.** Appendix A.    □

## 3. Adaptive α with PBIC Strategy

The Bayesian literature has been criticizing for several decades the implementation of hypothesis testing with fixed significance levels and, in particular, the use of the scale *p*-value < 0.05. An adaptive α allows us to adjust the statistical significance with the amount of information; see [5,11,12]. The adaptive values we work with in this section were calculated so that they allow to arrive to results equivalent to those obtained with a Bayes factor. In [5], the authors present an adaptive α based on BIC as
(5)$$\alpha_{n}\left(q\right)=\frac{{\left[\chi_{\alpha }^{2}\left(q\right)+q\log\left(n\right)\right]}^{\frac{q}{2}-1}}{2^{\frac{q}{2}-1}n^{\frac{q}{2}}\Gamma \left(\frac{q}{2}\right)}\times C_{\alpha },$$
where $C_{\alpha }$ is a calibration constant, and strategies for calculating it are presented in [5]. It yields a consistent procedure; it alleviates the problem of the divergence between practical and statistical significance; and it makes it possible to perform Bayesian testing by computing intervals with the calibrated α-levels.

An adaptive α is also presented in [6], but this time it is a version refined to nested linear models with calibration based on the Bayesian information criterion based on Prior PBIC [7],
(6)$$\alpha_{\left(b,n\right)}\left(q\right)=\frac{{\left[g_{n,\alpha }\left(q\right)+\log\left(b\right)+C\right]}^{\frac{q}{2}-1}}{b^{\frac{n-j}{2(n-1)}}\cdot {\left(\frac{2(n-1)}{n-j}\right)}^{q/2-1}\Gamma \left(\frac{q}{2}\right)}\times \exp\left\{-\frac{n-j}{2(n-1)}\left(g_{n,\alpha }\left(q\right)+C\right)\right\}.$$

Here, $b=\frac{|X_{j}^{t}X_{j}|}{|X_{i}^{t}X_{i}|}$ and $X_{i},X_{j}$ are design matrices and
$$C=2\sum_{m_{i}=1}^{q_{i}}\log\frac{\left(1-e^{-v_{m_{i}}}\right)}{\sqrt{2}v_{m_{i}}}-2\sum_{m_{j}=1}^{q_{j}}\log\frac{\left(1-e^{-v_{m_{j}}}\right)}{\sqrt{2}v_{m_{j}}},$$
$v_{m_{l}}=\frac{{\hat{\xi }}_{m_{l}}}{\left[d_{m_{l}}\left(1+n_{m_{l}}^{e}\right)\right]}$ with l=i,j corresponding to each model. Here, $n_{m_{l}}^{e}$, with l=i,j, refers to The Effective Sample Size (called TESS) corresponding to that parameter; see [7].

The adaptive α in (Equation 5) can also be presented using the PBIC strategy (this strategy was not considered in [5]), and the following expression is obtained
(7)$$\alpha_{n}\left(q\right)=\frac{{\left[\chi_{\alpha }^{2}\left(q\right)+q\log\left(n\right)+C\right]}^{\frac{q}{2}-1}}{n^{\frac{q}{2}}2^{\frac{q}{2}-1}\Gamma \left(\frac{q}{2}\right)}\times \exp\left\{-\frac{1}{2}\left(\chi_{\alpha }^{2}\left(q\right)+C\right)\right\}.$$
Note that this adaptive α is still of BIC structure, since the expression $\chi_{\alpha }^{2}\left(q\right)+q\log\left(n\right)$ remains.

### Example: Binomial Models

Consider comparing two binomial models $S_{1}∼\mathrm{binomial}\left(n_{1},p_{1}\right)$ and $S_{2}∼$
$\mathrm{binomial}(n_{2},p_{2})$ via the test
$$H_{0}:p_{1}=p_{2}\;\;\mathrm{vs}.\;\;H_{1}:p_{1}\neq p_{2}.$$

Defining $n=n_{1}+n_{2}$ and $\hat{p}$, the MLE from $p_{1}-p_{2}$, then (Equation 7) gives
(8)$$\alpha_{n}={\left[\frac{2}{n\pi (\chi_{\alpha }^{2}\left(1\right)+\log\left(n\right)+C)}\right]}^{1/2}\times \exp\left\{-\frac{1}{2}\left(\chi_{\alpha }^{2}\left(1\right)+C\right)\right\},$$
here, $\chi_{\alpha }^{2}\left(1\right)$ is the quantile α from chi-square with df=1, $C=-2\log\frac{\left(1-e^{-v}\right)}{\sqrt{2}v}$, $v={\hat{p}}^{2}/\left[d\left(1+n^{e}\right)\right]$, $d=\left(\frac{\sigma_{1}^{2}}{n_{1}}+\frac{\sigma_{2}^{2}}{n_{2}}\right),n^{e}=\max\left\{\frac{n_{1}^{2}}{\sigma_{1}^{2}},\frac{n_{2}^{2}}{\sigma_{2}^{2}}\right\}d$.

Table 1 shows the behavior of this adaptive $\alpha_{n}$ for α=0.05 and different values of $n_{1}$ and $n_{2}$.

## 4. Adjusting ${\mathrm{RLB}}_{\xi }$ Using Adaptive α

In this section, we combine (Equation 3) with the formulas for adaptive α in (Equation 6) and (Equation 7) for adjusting $RLB_{\xi }$ and obtaining an approximation to an objective Bayes factor. Indeed, we adjust the $RLB_{\xi }$ through the expression $B\left(\alpha \right)=B_{L}\left(\alpha ,\xi_{0}\right)\cdot g\left(\cdot \right)$, where *g* is determined in such a way that when B(α) is evaluated in (Equation 6) or (Equation 7), it converges to a constant (this allows us to obtain equivalent results from the Frequentist and Bayesian point of view, that is, the decision does not change).

Substituting *p* in (Equation 3) by the adaptive α value in (Equation 7) results in the following expression.
(9)$$B\left(\alpha ,q,n,\xi_{0}\right)=-\alpha^{\xi_{0}}\log\left(\alpha \right)\Gamma {\left(q/2\right)}^{\xi_{0}}n^{\frac{\xi_{0}q}{2}}{\left[\frac{2}{\chi_{\alpha }^{2}\left(q\right)+q\cdot \log\left(n\right)+C}\right]}^{\frac{\xi_{0}q}{2}-\left(\xi_{0}-1\right)}.$$

For a Uniform(0,1) *p*-value with $\xi_{0}=1$, this expression simplifies to
(10)$$B\left(\alpha ,q,n\right)=-\alpha \log\left(\alpha \right)\Gamma \left(q/2\right)n^{\frac{q}{2}}{\left[\frac{2}{\chi_{\alpha }^{2}\left(q\right)+q\cdot \log\left(n\right)+C}\right]}^{\frac{q}{2}}.$$

The refined version of this calibration for linear models is obtained when (Equation 3) is evaluated in (Equation 6)
(11)$$B\left(\alpha ,q,n,b\right)=-\alpha \log\left(\alpha \right)\Gamma \left(q/2\right)b^{\frac{n-j}{2(n-1)}}{\left[\frac{2(n-1)}{(g_{n,\alpha }\left(q\right)+\log\left(b\right)+C)\left(n-j\right)}\right]}^{\frac{q}{2}}$$
in this case, we only consider $\xi_{0}=1$.

### Balanced One-Way Anova

Suppose we have *k* groups with *r* observations each, for a total sample size of kr, and let $H_{0}:\mu_{1}=\cdots =\mu_{k}=\mu \;\;\mathrm{vs}.\;\;H_{1}:\mathrm{At}\mathrm{least}\mathrm{one}\;\mu_{i}\;\mathrm{different}$. Then, the design matrices for both models are
$$X_{1}=\left(\begin{array}{c} 1\\ 1\\ \vdots \\ 1 \end{array}\right)\;,X_{k}=\left(\begin{array}{cccc} 1 & 0 & \ldots & 0\\ 1 & 0 & \ldots & 0\\ \vdots & \vdots & \ldots & \vdots \\ 1 & 0 & \ldots & 0\\ 0 & 1 & \ldots & 0\\ 0 & 1 & \ldots & 0\\ \vdots & \vdots & \ldots & \vdots \\ 0 & 1 & \ldots & 0\\ \vdots & \vdots & \ldots & \vdots \\ 0 & 0 & \ldots & 1\\ 0 & 0 & \ldots & 1\\ \vdots & \vdots & \ldots & \vdots \\ 0 & 0 & \ldots & 1 \end{array}\right)\;,b=\frac{|X_{k}^{t}X_{k}|}{|X_{1}^{t}X_{1}|}=k^{-1}r^{k-1},$$
and the adaptive α for the linear model in accordance with what was presented in [6] is
$$\alpha \left(k,r\right)=\frac{{\left[g_{r,\alpha }\left(k-1\right)-\log\left(k\right)+\left(k-1\right)\log\left(r\right)+C\right]}^{\frac{k-3}{2}}}{{\left(k^{-1}r^{k-1}\right)}^{\frac{r-1}{2(r-1/k)}}{\left(\frac{2(r-1/k)}{r-1}\right)}^{\frac{k-3}{2}}\Gamma \left(\frac{k-1}{2}\right)}\times \exp\left\{-\frac{r-1}{2(r-1/k)}\left(g_{r,\alpha }\left(k-1\right)+C\right)\right\}.$$

Here, the number of replicas *r* is The Effective Sample Size (TESS). Therefore, the approximate Bayes factor for this test calculated with (Equation 8) is
$$B\left(\alpha ,k,r\right)=-\alpha \log\left(\alpha \right)\Gamma \left(\left(k-1\right)/2\right){\left(k^{-1}r^{k-1}\right)}^{\frac{r-1}{2(r-1/k)}}{\left[\frac{2(r-1/k)}{(g_{r,\alpha }\left(k-1\right)-\log\left(k\right)+\left(k-1\right)\log\left(r\right)+C)\left(r-1\right)}\right]}^{\frac{k-1}{2}}$$

A very important case arises when k=2. For this situation, the last formula simplifies to
(12)$$B\left(\alpha ,r\right)=-\alpha \log\left(\alpha \right){\left(\frac{r}{2}\right)}^{\frac{r-1}{2r-1}}{\left[\frac{2(r-1)\pi }{(g_{r,\alpha }\left(1\right)-\log\left(\frac{r}{2}\right)+C)\left(r-1\right)}\right]}^{\frac{1}{2}}$$

## 5. Obtaining Bounds for $P(H_{0}|\mathrm{Data})$

In this section, we use (Equation 9) and (Equation 11) to produce bounds for the posterior probability of the null hypothesis $H_{0}$.

Since for any Bayes factor $B_{01}$
$$B_{01}\geq B_{L}\left(p,\xi_{0}\right)\;\;\;\mathrm{with}\;\;\;\xi_{0}\geq 1,\;\mathrm{fixed}\;\mathrm{but}\;\mathrm{arbitrary},$$
a lower bound for the posterior probability of the null hypothesis can be obtained as
(13)$$\min P\left(H_{0}|Data\right)={\left[1+\frac{1}{B_{L}\left(p,\xi_{0}\right)}\right]}^{-1}.$$

Figure 2 shows these posterior probabilities (called $P_{RLB_{\xi_{0}}}$) for different values of $\xi_{0}$. To simplify the use of these Bayes factors, we call $BFG_{\xi_{0}}$ the Bayes factor of Equation (Equation 9), BFG the Bayes factor of Equation (Equation 10), and BFL the Bayes factor of Equation (Equation 11).

### 5.1. Testing Equality of Two Means

Consider comparing two normal means via the test
$$H_{0}:\mu_{1}=\mu_{2}\;\;\mathrm{versus}\;\;H_{1}:\mu_{1}\neq \mu_{2},$$
where the associated known variances, $\sigma_{1}^{2}$ and $\sigma_{2}^{2}$, are not equal.
$$Y=X\mu +\epsilon =\left(\begin{array}{cc} 1 & 0\\ \vdots & \vdots \\ 1 & 0\\ 0 & 1\\ \vdots & \vdots \\ 0 & 1 \end{array}\right)\left(\begin{array}{c} \mu_{1}\\ \mu_{2} \end{array}\right)+\left(\begin{array}{c} \epsilon_{11}\\ \vdots \\ \epsilon_{2n_{2}} \end{array}\right),$$
$$\times \epsilon ∼N(0,\mathrm{diag}\left\{\underset{n_{1}}{\underset{︸}{\sigma_{1}^{2},\cdots ,\sigma_{1}^{2}}},\underset{n_{2}}{\underset{︸}{\sigma_{2}^{2},\cdots ,\sigma_{2}^{2}}}\}\right)$$
Defining $\nu =(\mu_{1}+\mu_{2})/2$ and $\zeta =(\mu_{1}-\mu_{2})/2$ places this in the linear model comparison framework,
Y=Bνζ+ϵ
with
$$B=\left(\begin{array}{cc} 1 & 1\\ \vdots & \vdots \\ 1 & 1\\ 1 & -1\\ \vdots & \vdots \\ 1 & -1 \end{array}\right)$$
where we are comparing $M_{0}:\zeta =0$ versus $M_{1}:\zeta \neq 0$.

So, for BFG and BFL,
$$C=-2\log\frac{\left(1-e^{-v}\right)}{\sqrt{2}v}$$
$$v=\frac{{\hat{\zeta }}^{2}}{d(1+n^{e})},d=\left(\frac{\sigma_{1}^{2}}{n_{1}}+\frac{\sigma_{2}^{2}}{n_{2}}\right),n^{e}=\max\left\{\frac{n_{1}^{2}}{\sigma_{1}^{2}},\frac{n_{2}^{2}}{\sigma_{2}^{2}}\right\}\left(\frac{\sigma_{1}^{2}}{n_{1}}+\frac{\sigma_{2}^{2}}{n_{2}}\right).$$
A special case is the standard test of equality of means when $\sigma_{1}^{2}=\sigma_{2}^{2}=\sigma^{2}$. Then,
$$n^{e}=\min\left\{n_{1}\left(1+\frac{n_{1}}{n_{2}}\right),n_{2}\left(1+\frac{n_{2}}{n_{1}}\right)\right\}.$$

On the other hand, considering $\mu =\mu_{1}-\mu_{2}$ with $\sigma_{1}^{2}=\sigma_{2}^{2}=\sigma^{2}$:$H_{0}:\mu_{1}=\mu_{2}⟷\mu =0$;$H_{1}:\mu_{1}\neq \mu_{2}⟷\mu \neq 0$.
Assuming priors:$\mu |\sigma^{2},H_{1}∼Normal\left(0,\sigma^{2}/\tau_{0}\right),\tau_{0}\in \left(0,\infty \right)$;$\pi \left(\sigma^{2}\right)\propto 1/\sigma^{2}$ for both $H_{0}$ and $H_{1}$.
The Bayes factor is
(14)$$BF_{01}={\left(\frac{n+\tau_{0}}{\tau_{0}}\right)}^{1/2}{\left(\frac{t^{2}\frac{\tau_{0}}{n+\tau_{0}}+l}{t^{2}+l}\right)}^{\frac{l+1}{2}}$$
where
$$t=\frac{|\bar{Y}|}{s/\sqrt{n}}$$
a *t*-statistic with degrees of freedom l=n−1 and $n=n_{1}+n_{2}$; see [13].

Figure 3 shows the posterior probability for the null hypothesis $H_{0}$ when n=50 and n=100 for the Robust Lower Bound with $\xi_{0}=1$ (called $P_{RLB}$), the Bayes factor BFL (called $P_{BFL}$), the Bayes factor BFG (called $P_{BFG}$), and the Bayes factor $BF_{01}$ (called $P_{BF_{01}}$). Note that the posterior probability with $BF_{01}$ when $\tau_{0}=6$ looks very similar to the result obtained using the Bayes factors BFL and BFG.

We now present a simulation that shows how our adjustment, or calibration, to $RLB_{\xi }$ works quite similarly to an exact Bayes factor. We perform the following experiment: We simulate *r* data points from each of the two normal distributions, $N(\mu_{1},\sigma )$ and $N(\mu_{2},\sigma )$. We reproduce this *K* times. For all *K* simulations, $\mu_{1}-\mu_{2}=0$. For all *K* replicates, we test the hypotheses $H_{0}:\mu_{1}=\mu_{2}$ vs. $H_{1}:\mu_{1}\neq \mu_{2}$, and then we count how many of the *p*-values lie between 0.05−ε and 0.05. Note that all of these *p*-values would be considered sufficient to reject $H_{0}$ if α=0.05 is selected. Finally, we determine the proportion of these “significant” *p*-values obtained from samples where $H_{0}$ is true.

Table 2 presents the mean percentage of these significant *p*-values coming from samples, where $H_{0}$ is true for 100 iterations of the simulation scheme with K=8000, σ=1, and ε=0.05 for r=10,50,100,500, and 1000. As expected, the distribution of the *p*-values behaved Uniform(0,1) under $H_{0}$, since $H_{0}$ was assumed true in the *K* replicates. Table 2 also presents the proportion of posterior probability of $H_{0}$ greater than or equal to 0.5 (50%) when using the $RLB_{\xi }$, when corrected according to the method suggested in this document (Equations (Equation 10) and (Equation 11)), and when an exact Bayes factor (Equation (Equation 14)) is used. It is clear that the method suggested here behaves very similarly to an exact Bayes factor.

### 5.2. Fisher’s Exact Test

This is an example where the *p*-value is a pseudo-*p*-value (see the example 8.3.30 in [4]). Let $S_{1}$ and $S_{2}$ be independent observations with $S_{1}∼\mathrm{binomial}\left(n_{1},p_{1}\right)$ and $S_{2}∼\mathrm{binomial}\left(n_{2},p_{2}\right)$. Consider testing $H_{0}:p_{1}=p_{2}$ vs. $H_{1}:p_{1}\neq p_{2}$.

Under $H_{0}$, if we let *p* be the common value of $p_{1}=p_{2}$, the joint pmf of $\left(S_{1},S_{2}\right)$ is
f(s1,s2|p)=n1s1n2s2ps1+s2(1−p)n1+n2−(s1+s2)
and the conditional pseudo-*p*-value is
(15)$$p\left(s_{1},s_{2}\right)=\sum_{j=s_{1}}^{\min\{n_{1},s\}}f\left(j|s\right),$$
the sum of hypergeometric probabilities, with $s=s_{1}+s_{2}$.

**Remark 2.** 
*It does not seem to be simple to estimate the appropriate $\xi_{0}$ that best fits the pseudo-p-value in (Equation 15), in Figure 4 some arbitrary possibilities are given.*


It is important to note that in Bayesian tests with a point null hypothesis, it is not possible to use continuous prior densities, because these distributions (as well as posterior distributions) will grant zero probability to $p=(p_{1}=p_{2})$. A reasonable approximation will be to give $p=(p_{1}=p_{2})$, a positive probability $\pi_{0}$, and to $p\neq (p_{1}=p_{2})$ the prior distribution $\pi_{1}g_{1}\left(p\right)$, where $\pi_{1}=1-\pi_{0}$ and $g_{1}$ proper. One can think of $\pi_{0}$ as the mass that would be assigned to the real null hypothesis, $H_{0}:p\in \left(\left(p_{1}=p_{2}\right)-b,\left(p_{1}=p_{2}\right)+b\right)$ if it had not been preferred to approximate by the null point hypothesis. Therefore, if
$$\pi \left(p\right)=\left\{\begin{array}{cc} \pi_{0} & p=(p_{1}=p_{2})\\ \pi_{1}g_{1}\left(p\right) & p\neq (p_{1}=p_{2}) \end{array}\right.$$
then
$$\begin{array}{ccc} m(s) & = & \int_{\Theta }f\left(s|p\right)\pi \left(p\right)dp\\ & = & f\left(s|\left(p_{1}=p_{2}\right)\right)\pi_{0}+\pi_{1}\int_{p\neq (p_{1}=p_{2})}f\left(s|p\right)g_{1}\left(p\right)dp\\ & = & f\left(s|\left(p_{1}=p_{2}\right)\right)\pi_{0}+\left(1-\pi_{0}\right)m_{1}\left(s\right) \end{array}$$
where $m_{1}\left(s\right)=\int_{p\neq (p_{1}=p_{2})}f\left(s|p\right)g_{1}\left(p\right)dp$ is the marginal density of $\left(S=S_{1}+S_{2}\right)$ with respect to $g_{1}$.

So,
$$\pi \left(\left(p_{1}=p_{2}\right)|s\right)=\frac{\pi_{0}f\left(s|\left(p_{1}=p_{2}\right)\right)}{m(s)}$$
thus
$$\begin{array}{ccc} \mathrm{posterior}\;\mathrm{odds} & = & \frac{\pi (\left(p_{1}=p_{2}\right)|s)}{1-\pi (\left(p_{1}=p_{2}\right)|s)}\\ & = & \frac{f\left(s|\left(p_{1}=p_{2}\right)\right)\pi_{0}}{m\left(s\right)\left(1-\frac{f\left(s|\left(p_{1}=p_{2}\right)\right)\pi_{0}}{m(s)}\right)}\\ & = & \frac{f\left(s|\left(p_{1}=p_{2}\right)\right)\pi_{0}}{m\left(s\right)-f\left(s|\left(p_{1}=p_{2}\right)\right)\pi_{0}}\\ & = & \frac{f\left(s|\left(p_{1}=p_{2}\right)\right)\pi_{0}}{\left(1-\pi_{0}\right)m_{1}\left(s\right)}\\ & = & \frac{\pi_{0}f\left(s|\left(p_{1}=p_{2}\right)\right)}{\pi_{1}m_{1}\left(s\right)}\\ & = & \mathrm{prior}\;\mathrm{odds}\cdot \frac{f(s|\left(p_{1}=p_{2}\right))}{m_{1}\left(s\right)} \end{array}$$
and the Bayes factor is
$$B_{01}=\frac{f(s|\left(p_{1}=p_{2}\right))}{m_{1}\left(s\right)}.$$

Now, if we take $g_{1}\left(p\right)=\mathrm{Beta}\left(a,b\right)$ such that $E\left(p\right)=\frac{a}{a+b}=\left(p_{1}=p_{2}\right)$, then
$$BF_{Test}=\frac{B(a,b)}{B(s+a,n_{1}+n_{2}-s+b)}p^{s}{\left(1-p\right)}^{n_{1}+n_{2}-s}.$$

Figure 4 shows the posterior probability for the null hypothesis $H_{0}$ when $n=n_{1}+n_{2}=50$ and 100, for the Robust Lower Bound, the Bayes factor $BFG_{\xi_{0}}$ (called $P_{BFG_{\xi_{0}}}$), the Bayes factor BFG (called $P_{BFG}$), and the Bayes factor $BF_{Test}$ (called $P_{BF_{Test}}$). We can note that all the $P_{BFG_{\xi_{0}}}$ are comparable, even though in the case $\xi_{0}=1$ ($P_{BFG}$) it is a *p*-value and not a pseudo-*p*-value.

### 5.3. Linear Regression Models

Consider comparing two nested linear models $M_{3}:y_{l}=\lambda_{1}+\lambda_{2}x_{l2}+\lambda_{3}x_{l3}+\epsilon_{l}$ with $M_{2}:y_{l}=\lambda_{1}+\lambda_{2}x_{l2}+\epsilon_{l}$ via the test
$$H_{0}:M_{2}\;\;\mathrm{versus}\;\;H_{1}:M_{3},$$
with 1≤l≤n, and the errors $\epsilon_{l}$ are assumed to be independent and normally distributed with unknown residual variance $\sigma^{2}$. According to the Equation (Equation 3) in [6,7]
$$b=\left(n-1\right)s_{3}^{2}\left(1-\rho_{23}^{2}\right),$$
where $s_{3}^{2}$ is the variance $x_{v3}$, $\rho_{23}$ is the correlation between $x_{v2}$ and $x_{v3}$, and
$$C=2\log\frac{\left(1-e^{-v_{2}}\right)}{\sqrt{2}v_{2}}-2\log\frac{\left(1-e^{-v_{3}}\right)}{\sqrt{2}v_{3}},$$
where $v_{2}={\hat{\lambda }}_{2}^{2}/\left[d_{2}\left(1+n_{2}^{e}\right)\right]$, $d_{2}=\sigma^{2}/s_{x_{l2}}^{2}$, $n_{2}^{e}=s_{x_{l2}}^{2}/\max_{i}\left\{{\left(x_{i2}-{\bar{x}}_{2}\right)}^{2}\right\}$ and $v_{3}={\hat{\lambda }}_{3}^{2}/$$\left[d_{3}\left(1+n_{3}^{e}\right)\right]$, $d_{3}=\sigma^{2}{\left({\tilde{X}}^{t}\tilde{X}\right)}^{-1}$, $n_{3}^{e}={\tilde{X}}^{t}\tilde{X}/\max_{i}\left\{\right|{\tilde{X}}_{i}{|}^{2}\}$ with $\tilde{X}=\left(I_{n}-X^{*}{\left(X^{*t}X^{*}\right)}^{-1}X^{*}\right)x_{l3}$ and $X^{*}=\left(1_{n}|x_{l2}\right)$.

As an example, we analyze a data set taken from [14], which can be accessed at http://academic.uprm.edu/eacuna/datos.html (accessed on 13 January 2022). We want to predict the average mileage per gallon (denoted by mpg) of a set of n=82 vehicles using four possible predictor variables: cabin capacity in cubic feet (vol), engine power (hp), maximum speed in miles per hour (sp), and vehicle weight in hundreds of pounds (wt).

Through the Bayes factors BFG and BFL, we want to choose the best model to predict the average mileage per gallon by calculating the posterior probability of the null hypothesis of the following test   
$$H_{0}:M_{2}:\mathrm{mpg}\;=\;\lambda_{1}+\lambda_{2}{\mathrm{wt}}_{l}+\epsilon_{l}\;\mathrm{vs}.\;H_{1}:M_{3}:\mathrm{mpg}\;=\;\lambda_{1}+\lambda_{2}{\mathrm{wt}}_{l}+\lambda_{3}{\mathrm{sp}}_{l}+\epsilon_{l}$$
with α=0.05, q=1, j=3, the posterior probabilities for the null hypothesis $H_{0}$ are
$$P_{BFL}=0.9253192,P_{BFG}=0.7209449.$$
The use of this posterior probability in both cases will change the inference, since the *p*-value of the F test is p=0.0325, which is smaller than 0.05.

#### Findley’s Counterexample

Consider the following simple linear model [15]
$$Y_{i}=\frac{1}{\sqrt{i}}\cdot \theta +\epsilon_{i},\;\;\mathrm{where}\;\epsilon_{i}∼N\left(0,1\right),i=1,2,3,\ldots ,n$$
and we are comparing the models $H_{0}:\theta =0$ and $H_{1}:\theta \neq 0$. This is a classical and challenging counterexample against BIC and the Principle of Parsimony. In [7], the inconsistency of BIC is shown, but the consistency of PBIC is shown in this problem.

Here, we show through the posterior probabilities of the null hypothesis that the Bayes factor BFG ( based on BIC) is inconsistent, while the Bayes factor BFL ( based on PBIC) is consistent if it is. We perform the analysis in two contexts: First, when *n* grows and α=0.05 or α=0.01 are fixed. Second, when *n* is fixed and 0<α<0.05. For calculations
$$C=-2\log\frac{\left(1-e^{-v}\right)}{\sqrt{2}v},v=\frac{{\hat{\theta }}^{2}}{d(1+n^{e})},d={\left(\sum_{i=1}^{n}\frac{1}{i}\right)}^{-1},n^{e}=\sum_{i=1}^{n}\frac{1}{i}.$$

Figure 5 and Figure 6 show, through the posterior probability of the null hypothesis $H_{0}$, the consistency of the Bayes factor based in PBIC ($P_{BFL}$), as well as the inconsistency of the Bayes factor based in BIC ($P_{BFG}$).

## 6. Discussion and Final Comments

1.Lower bounds have been an important development to give practitioners alternatives to classical testing with fixed α levels. A deep-seated problem with the useful bound −e·p·log(p) is that it depends on the *p*-value, which it should, but it is static, not a function of the sample size *n*. This limitation makes the bound of little use for moderate to large sample sizes, where it is arguably the correction to *p*-values more needed.2.The approximation develops here as a function of *p*-values, and sample size has a distinct advantage over other approximations, such as BIC, in that it is a valid approximation for any sample size.3.The (approximate) Bayes factors (Equation 9) and (Equation 11) are simple to use and provide results equivalent to the sensitive *p*-value Bayes factors of hypothesis tests. In this article, we extended the validity of the approximation for “pseudo-*p*-values,” which are ubiquitous in statistical practice. We hope that this development will give tools to the practice of statistics to make the posterior probability of hypotheses closer to everyday statistical practice, on which *p*-values (or pseudo-*p*-values) are calculated routinely. This allows an immediate and useful comparison between raw-*p*-values and (approximate) posterior odds.

## Figures and Tables

**Figure 1 entropy-25-00618-f001:**
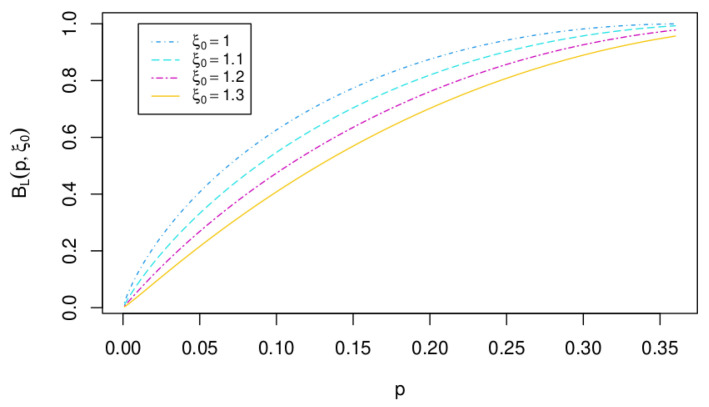
Extended Robust Lower Bound $RLB_{\xi_{0}}$ as a function of *p* for different values of $\xi_{0}$.

**Figure 2 entropy-25-00618-f002:**
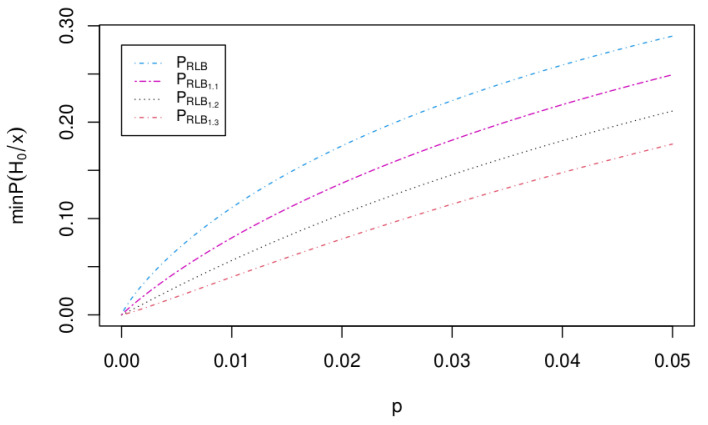
Lower bound for posterior probability for the null hypothesis $H_{0}$ (in (Equation 13)) for $\xi_{0}=1$, $\xi_{0}=1.1,\xi_{0}=1.2,\xi_{0}=1.3$.

**Figure 3 entropy-25-00618-f003:**
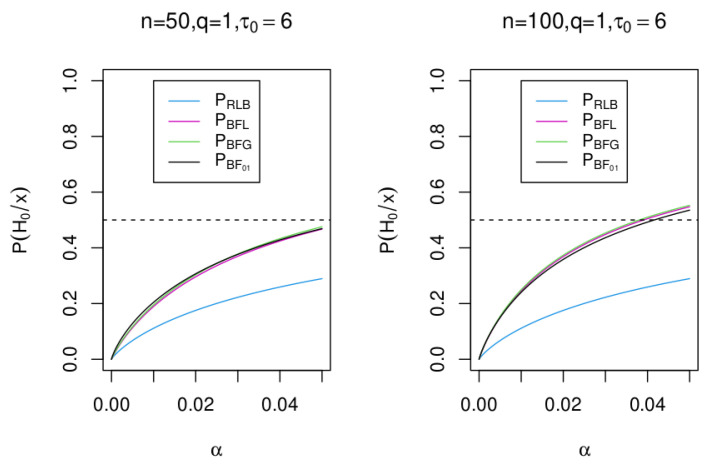
Posterior probability for the null hypothesis $H_{0}$ for n=50 and n=100 using the Bayes factor $RLB_{\xi_{0}}$ with $\xi_{0}=1$, the Bayes factor $BF_{01}$, and the Bayes factor BFL and BFG.

**Figure 4 entropy-25-00618-f004:**
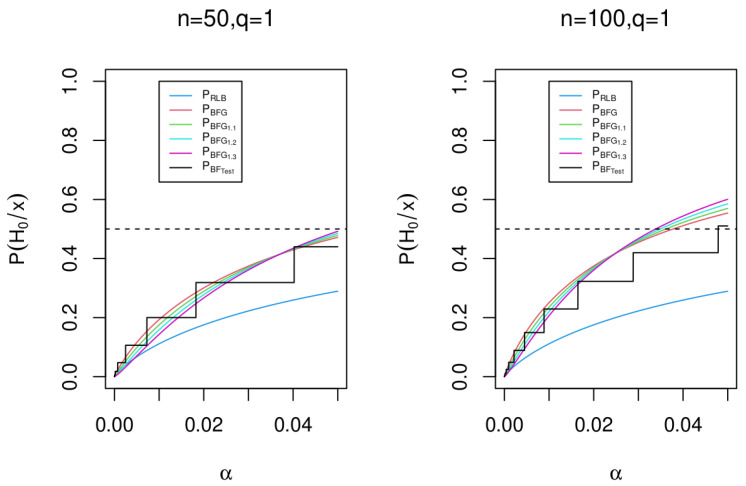
Posterior probability for the null hypothesis $H_{0}$ for n=50 and n=100 using the Bayes factor $RLB_{\xi_{0}}$ with $\xi_{0}=1$, the Bayes factor $BF_{Test}$, the Bayes factor $BFG_{\xi_{0}}$, and the Bayes factor BFG.

**Figure 5 entropy-25-00618-f005:**
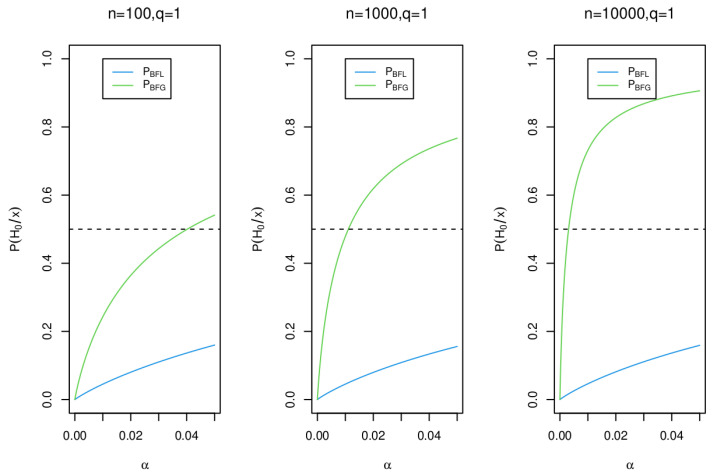
Posterior probability for the null hypothesis $H_{0}$ for n=100, n=1000 and n=10,000 using the Bayes factors BFL and BFG.

**Figure 6 entropy-25-00618-f006:**
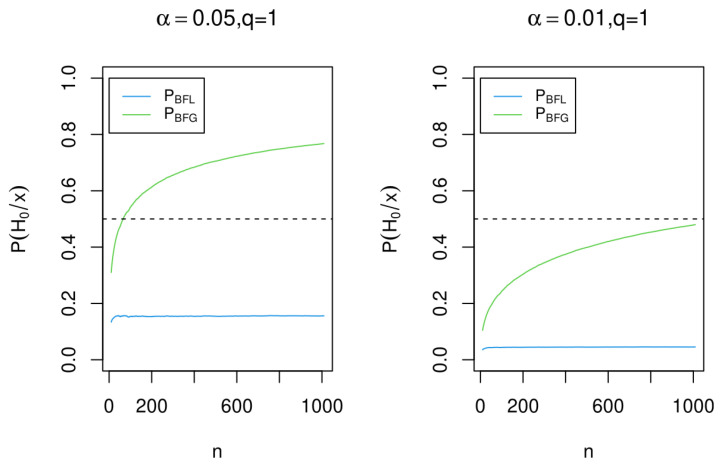
Posterior probability for the null hypothesis $H_{0}$ for α=0.05 and α=0.01 using the Bayes factors BFL and BFG when *n* grows.

**Table 1 entropy-25-00618-t001:** Adaptive α via PBIC in (Equation 8) for testing equality of two proportions for different sample sizes when α=0.05.

		Adaptive α via PBIC ($\alpha_{n}$)
$n_{1}$	$n_{2}$	$n=n_{1}+n_{2}$
10	10	0.0068
25	25	0.0040
50	50	0.0027
100	50	0.0021
50	100	0.0021
100	100	0.0018

**Table 2 entropy-25-00618-t002:** Mean percentage of *p*-values less than 0.05 (considered significant) coming from data generated under the null hypothesis for 100 experiments, where K=8000 testing problems are generated under $H_{0}:\mu_{1}=\mu_{2}$. This experiment is performed for different groups with sample sizes *r*. Corrected and uncorrected Bayes factors are considered, as well as an exact Bayes factor.

		% Of Samples with $P(H_{0}|x)\geq 0.5$			
* **r** *	**% Of Samples with** p<0.05	${\mathrm{RLB}}_{\xi }$	BFG	BFL	${\mathrm{BF}}_{01}$
10	5%	0%	58%	66%	75%
50	5%	0%	81%	86%	87%
100	5%	0%	86%	89%	91%
500	5%	0%	94%	96%	96%
1000	5%	0%	95%	96%	97%

## Data Availability

The real datasets are freely available in http://academic.uprm.edu/eacuna/datos.html.

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
