# Peer review of "From p-Values to Posterior Probabilities of Null Hypotheses"

_entropy, 2023, doi:10.3390/e25040618_

Round 1

Reviewer 1 Report

I have made several comments and suggestions in a PDF file that I attached here, but overall the article is quite good and I've recommended publication after minor revisions. 

Author Response

Attach

Reviewer 2 Report

I am writing to provide my review of the manuscript titled "From p-Values to Posterior Probabilities of Null Hypothesis ", which I had the privilege of reading and analyzing.

Overall, I find the research to be of good quality and well-written. The paper presents a comprehensive analysis of the research question and provides clear and concise explanations of the methods used and the results obtained. The literature review is also thorough and provides a solid foundation for the study.

However, I noticed that the paper did not include any digital simulation. I believe that a digital simulation can provide a valuable contribution to the study by providing a more robust analysis and validating the results obtained. Therefore, I recommend that the authors add a digital simulation to their research.

In addition, I recommend that the authors provide the codes used in their analysis. By providing the codes, other researchers can replicate the study and build upon it. This would increase the impact and relevance of the research in the field.

Overall, I recommend that the paper be accepted for publication after minor revisions, including the addition of a digital simulation and the provision of the codes used in the analysis.

Thank you for the opportunity to review this paper, and I look forward to reading the revised version

Author Response

Attach

Reviewer 3 Report

Title: From p-Values to Posterior Probabilities of Null Hypothesis

By: Daiver Vélez Ramos, , Luis R. Pericchi Guerra, and María Eglée Pérez Hernández

Submitted to: Entropy,   Ms I.d. Entropy-2254425

Report     03/03/2023

The authors propose a method to connect the Bayesian posterior probability to the frequentist's p-value or pseudo-p-value, which is an improvement of the existing method. Their method is illustrated by several examples. 

Major Comments:

* The proposed method can take into consideration of sample size, and hence provides some improvement over existing methods. The method is based on some other sample size based adjustment methods, such as [8], [11], [2]. It is not clear from the presentation of the paper, which part is new from the authors.

* There is lack of motivation/rationale for the proposed methods. For example, in Section 3, the adaptive \alpha is just given by expression (4), without motivation/rationale and derivation.  

* There is no simulation study to examine the finite sample size performance of the proposed method, and compare with some commonly used existing methods, which is a necessary part for most statistical journals.

Author Response

Attach

Round 2

Reviewer 1 Report

I want to make it clear that when I rated some of the aspects of the article "average," I meant average among articles that deserve to be published.  That is, it is not a criticism or a negative comment.  It's actually positive. 

There are a few things that I would express slightly differently in English in this revised version of the article, but I don't think I need to correct them, because some of them end up being just differences of style, and I don't think any of them get in the way of readers understanding the article. 

The points I made in the first referee report were addressed in the reply and in the new version of the article.  I find the article much clearer now. 

The only suggestions I will make now are two very simple ones, more editing than content, but at least the first of the two is possibly more appropriate for the authors to change than for the editors to change. 

The first is a simple replacement of a Spanish word with the corresponding English word.  Just after line 123 of the new version of the article, there is an occurrence of "con" (the Spanish word that corresponds to the English word "with") in the indented mathematical expression.  I suggest replacing that with the English word "with."  It currently reads "B01 BL(p, ξ0) con ξ0 1, fixed but arbitrary"

The other is that just after the last reference in the article, part of a reference is there (". , 25, 19. https://doi.org/10.3390/e25010019").  That appears to be left over from the editing to include reference 8.  It's easy to remove. 

I'll put both of these trivial suggestions in comments to the editors as well. 

I have selected the option "Accept after minor revision (corrections to minor methodological errors and text editing)" on the reviewer form, but with these two trivial changes made, I would change that to "Accept in present form." 

I believe I have selected the correct option ("Accept after minor revision") because I see text on the reviewer form that says "Please note that a recommendation of "Minor revisions" implies that the manuscript can be accepted without the reviewer's further review. The Academic Editors will assess the revisions made by the authors."

Author Response

attach

Reviewer 3 Report

Title: From p-Values to Posterior Probabilities of Null Hypothesis

By: Daiver Vélez Ramos, , Luis R. Pericchi Guerra, and María Eglée Pérez Hernández

Submitted to: Entropy,   Ms I.d. Entropy-2254425

Report     03/03/2023

The authors propose a method to connect the Bayesian posterior probability to the 

frequentist's p-value or pseudo-p-value, which is an improvement of the existing

method. Their method is illustrated by several examples. 

Major Comments:

* The proposed method can take into consideration of sample size, and hence provides

  some improvement over existing methods. The method is based on some other sample size

  based adjustment methods, such as [8], [11], [2]. After the reviosn, it is still not 

  easy to see which part is new from the authors. The authors should use some 

  subsections with title such as "3.1. The existing method", "3.2. The proposed 

  adaptive p-value", ... etc. to make this clear, not just mix them in the text with 

  other existing methods. 

* There is no simulation study to examine the finite sample size performance of the

  proposed method, and compare with some commonly used existing methods, which is a

  necessary part for most statistical journals.

Minor Comments.

* All proofs should be put into an Appendix.

* There is no need to present the computation/figure codes in the Appendix.

Author Response

attach

Round 3

Reviewer 3 Report

Most of my previous comments are addressed, I have no more comment.